# Fenofibrate Facilitates Post-Active Tuberculosis Infection in Macrophages and is Associated with Higher Mortality in Patients under Long-Term Treatment

**DOI:** 10.3390/jcm9020337

**Published:** 2020-01-25

**Authors:** Ching-Lung Liu, Yen-Ta Lu, I-Fan Tsai, Ling-Chiao Wu, Wu-Chien Chien, Chi-Hsiang Chung, Kuo-Hsing Ma

**Affiliations:** 1Division of Chest Medicine, Department of Internal Medicine, MacKay Memorial Hospital, Taipei 104, Taiwan; clliu.5839@gmail.com (C.-L.L.); ytlhl@mmh.org.tw (Y.-T.L.); 2Department of Medicine, MacKay Medical College, New Taipei City 252, Taiwan; 3Graduate Institute of Medical Sciences, National Defense Medical Center, Taipei 114, Taiwan; 4Department of Medical Research, MacKay Memorial Hospital, Taipei 104, Taiwan; m9209001@gmail.com (I.-F.T.); macototamiya@hotmail.com (L.-C.W.); 5Department of Medical Research, National Defense Medical Center, Taipei 114, Taiwan; chienwu@mail.ndmctsgh.edu.tw; 6School of Public Health, National Defense Medical Center, Taipei 114, Taiwan; g694810042@gmail.com; 7Department of Biology and Anatomy, National Defense Medical Center, Taipei 114, Taiwan

**Keywords:** fenofibrate, tuberculosis infection, lipid, mycobacterial tuberculosis

## Abstract

Background: *Mycobacterium tuberculosis* (*Mtb*) is an intracellular pathogen that infects and persists in macrophages. This study aimed to investigate the effects of long-term fenofibrate treatment in patients with tuberculosis (TB), and the intracellular viability of *Mtb* in human macrophages. Methods: Epidemiological data from the National Health Insurance Research Database of Taiwan were used to present outcomes of TB patients treated with fenofibrate. In the laboratory, we assessed *Mtb* infection in macrophages treated with or without fenofibrate. *Mtb* growth, lipid accumulation in macrophages, and expression of transcriptional genes were examined. Results: During 11 years of follow-up, TB patients treated with fenofibrate presented a higher risk of mortality. Longer duration of fenofibrate use was associated with a significantly higher risk of mortality. Treatment with fenofibrate significantly increased the number of bacilli in human macrophages in vitro. Fenofibrate did not reduce, but induced an increasing trend in the intracellular lipid content of macrophages. In addition, dormant genes of *Mtb*, *icl1*, *tgs1*, and *devR*, were markedly upregulated in response to fenofibrate treatment. Our results suggest that fenofibrate may facilitate intracellular *Mtb* persistence. Conclusions: Our data shows that long-term treatment with fenofibrate in TB patients is associated with a higher mortality. The underlying mechanisms may partly be explained by the upregulation of *Mtb* genes involved in lipid metabolism, enhanced intracellular growth of *Mtb*, and the ability of *Mtb* to sustain a nutrient-rich reservoir in human macrophages, observed during treatment with fenofibrate.

## 1. Introduction

Tuberculosis (TB), caused by *Mycobacterium tuberculosis* (*Mtb*), is a key health problem and a disease affecting millions of people worldwide [1]. In 2018, there were approximately 10 million cases of TB and 1.2 million deaths caused by TB [1]. In Taiwan, even though the incidence of TB has dropped from 63.2 (in 2007) to 41.4 (in 2017) per 100,000 people, it still presents the highest incidence and mortality rate [2]. This disease is transmitted via inhalation of aerosolized bacilli, which then replicate within alveolar macrophages in lung tissue. In the host, immune cells can be recruited to the affected sites, and encase the invading mycobacteria, forming a structure known as the granuloma [3,4]. A granuloma is composed of infected alveolar macrophages, monocytes, multinucleated giant cells, epithelioid cells, and most notably, foamy macrophages [4,5]. The foamy aspect of these cells results from the accumulation of neutral lipids, typically triacylglycerides and esterified and non-esterified sterols. These foamy macrophages are nutrient-rich reservoirs for *Mtb* persistence [6,7].

Published data support that regulating cholesterol in macrophages may affect the intracellular growth of *Mtb* [8,9]. However, questions remain regarding the use of hypolipidemic therapy for the treatment of *Mtb* infection [10]. By the activation of peroxisome proliferator activated receptor α (PPARα), mainly expressed in the liver, heart, monocytes, and macrophages, fenofibrate increases lipolysis and elimination of lipids from plasma [11,12,13]. It is commonly prescribed for the treatment of hypertriglyceridemia or mixed hyperlipidemia [13,14,15]. This study aimed to investigate epidemiological data to obtain outcomes of TB patients treated with fenofibrate, and to examine the effect of fenofibrate and lipids on the intracellular viability of *Mtb* in human macrophages.

## 2. Materials and Methods

### 2.1. Epidemiological Survey of Outcomes of TB Patients Treated with Fenofibrate

We used the Longitudinal Health Insurance Database, which includes 2,000,000 beneficiaries randomly sampled from the Registry of the National Health Insurance Research Database (NHIRD) (provided by the Health and Welfare Statistics Application Center, Ministry of Health and Welfare) between 1 January 2000 and 31 December 2010. The database contains all relevant information about the catastrophic illness status, including diagnostic codes based on the International Classification of Disease—Clinical Modification, Ninth Revision (ICD-9-CM). Patients with newly onset TB (ICD-9-CM: 011.x) and those who had ever been prescribed fenofibrate in the year 2000 were eligible for inclusion in this study. The date of the initial TB diagnosis was used as the date of cohort enrollment. TB patients without any use of fenofibrate therapy were recruited for the comparator cohort. We applied propensity score matching (gender, age, and index date) at a ratio of 1:4. All patients were followed up until death or until 31 December 2010. Cumulative defined daily dose (cDDD) represented the total dose of drug prescribed during study period. The outcome was all-cause death during study period.

### 2.2. Cell Lines and Culture

Human monocytes were isolated from white blood cell concentrates of healthy human volunteers (Taiwan Blood Service Foundation, Taipei, Taiwan). Macrophages were differentiated from monocytes after 6 d of culture in RPMI 1640 medium (GIBCO/Life Technologies, Grand Island, NY, USA) with 10% fetal bovine serum (FBS) (Biological Industries, Haemek, Israel) in the presence of 50 units/mL recombinant human GM-CSF (Peprotech Ltd., NJ, USA).

### 2.3. Mycobacterium Tuberculosis Culture

*Mtb* were obtained from a clinical virulent strain. *Mtb* was maintained in 7H9 broth supplemented with 0.2% glycerol, 0.05% Tween-80, and 10% oleic acid-albumin-dextrose-catalase enrichment (Difco, Becton Dickinson and Company, Sparks, MD, USA) to an optical density at 600 nm of 0.3.

### 2.4. Macrophage Infection and Mtb Colony-Forming Units

Macrophages were plated in 12-well plates and infected with *Mtb* for 2 h. Extracellular bacilli were removed by washing macrophages with phosphate buffered saline (PBS). The cells were treated with fenofibrate (50 µM, LKT Laboratories, MN, USA) in the culture medium. At the indicated periods, cells were lysed with 0.01% SDS solution. To generate quantitative cultures for *Mtb*, a 10-fold serial dilution was performed, followed by inoculation of 50 μL triplicate samples on 7H11 agar plates (Creative Microbiologicals, Ltd., Taipei County, Taiwan) for 3 weeks. The resulting growth of *Mtb* was reported as mean colony-forming units (CFUs) per milliliter.

### 2.5. Oil Red O Staining

Fixed macrophages were rinsed with 60% isopropanol and incubated with filtered Oil Red O (ORO) working solution (3:2 ratio; 3 mg/mL ORO stock solution: water) for 30 min. After removing the ORO working solution, cells were incubated with hematoxylin for 1 min and washed with water. The slides were observed under an inverted microscope.

### 2.6. Nile Red Staining

Macrophages were treated with fenofibrate (50 μM) with or without human very low-density lipoprotein (VLDL) (50 µg/mL, Calbiochem, San Diego) for 24 h. Cells were then infected with *Mtb* (multiplicity of infection, MOI = 5) for 2 h at 37 °C. At the indicated time points, macrophages were washed repeatedly to remove extracellular bacteria and cultured in RPMI 1640 medium with 10% FBS. Afterwards, cells were detached from the culture plate using a lidocaine/EDTA solution (5 mM EDTA and 4 mg/mL lidocaine in PBS pH 7.2). Nile red (100 µg/mL, Sigma-Aldrich Co., St. Louis, MO, USA) was added directly to the cells in a 1:100 dilution for 10 min. The mean fluorescence intensity of stained cells was measured using a FACSCalibur flow cytometer and analyzed using CellQuest software (BD Bioscience, San Jose, CA, USA).

### 2.7. Triglyceride colorimetric Assay

Differentiated macrophages (1 × 10^7^) were treated with fenofibrate (50 μM) with or without human VLDL (50 µg/mL) for 48 h. The cells were detached using 1% Triton-X 100 solution. The cell suspension was centrifuged at 10,000× *g* for 10 min at 4 °C. Cayman’s triglyceride colorimetric assay enzyme buffer (*Cayman* Europe, Tallinn, Estonia) was added to the supernatant and it was incubated for 15 min at 25 °C. Absorbance was detected at 550 nm using a plate reader.

### 2.8. RNA Isolation and Quantitative Reverse Transcription PCR (qRT-PCR)

Infected macrophages were lysed with RLT buffer and transferred to tubes containing 0.1-mm-diameter zirconia/silica beads (Biospec, Bartlesville, OK, USA). Mycobacteria were disrupted using a bead beater, and then centrifuged for 1 min at 15,000× *g*. RNA in the supernatant was then column-purified using RNeasy plus mini kit (Qiagen, Valencia, CA, USA) according to the manufacturer’s instructions. RNA was reverse-transcribed using random primers and Superscript III reverse transcriptase (Life Technologies, Grand Island, NY, USA). Real-time PCR was performed on cDNA using KAPA FAST SYBR green assay (reverse and forward primers of the genes RV0467, RV3130, RV3133 were used) [16,17,18]. Fluorescence was measured by ABIPrism 7500 (Applied Biosystems/Life Technologies, Grand Island, NY, USA).

### 2.9. Statistical Analysis

Differences in the risk of all-cause mortality between the study and comparison cohorts were estimated using the Kaplan–Meier method with the log-rank test. Laboratory data are reported as the means ± SEMs. Unpaired *t-*test and Mann–Whitney *U*-test were used for analysis. Statistical analysis was performed using the Prism 3.0 software (GraphPad Software Inc., San Diego, CA, USA). A *p* value of <0.05 was considered statistically significant.

## 3. Ethics Statement

This study was approved by the Institutional Review Board of the MacKay Memorial Hospital in Taipei, Taiwan (09MMHIS180).

## 4. Results

### 4.1. Fenofibrate Therapy is Associated with Adverse Outcomes in Patients with TB

The study cohort comprised 605 individuals who had been diagnosed with TB and received fenofibrate in the year 2000. Another 2420 TB subjects not treated with fenofibrate were randomly selected as the comparison cohort (Figure 1a) (Appendix A). There were 108 deaths (17.9%) in the study cohort, and 354 (14.6%) in the comparison cohort of patients followed-up for 11 years. The causes of death in each group of TB infected patients were similar (with versus without fenofibrate, 7.4% versus 5.4%) (Appendix A). However, the risk of death was higher in the TB patients who were treated with fenofibrate (log-rank *p* values = 0.035) (Figure 1b) (Appendix A). A longer duration of fenofibrate use (>1 year) was found to be associated with an increasing risk of mortality (log-rank *p* values < 0.001) (Figure 1c).

### 4.2. The Effect of Fenofibrate on the Intracellular Lipid Content of Human Macrophages

Next, we sought to examine the underlying mechanisms whereby the use of fenofibrate might be leading to a higher mortality in an in vitro model. Human macrophages were treated with fenofibrate with or without VLDL. The cells were then stained with Oil Red O, and that was followed by light microscopy examination (Figure 2a). In order to quantify the content of intracellular lipid, the cells infected with mycobacteria were also examined by a lipid-specific dye Nile Red and then quantified by flow cytometry (Figure 2b,c). Our data showed the intracellular lipid accumulation did not decrease in response to fenofibrate treatment with/without VLDL. Similarly, the level of intracellular triglycerides in human macrophages did not decrease in the fenofibrate group, although it showed a trend towards an increase in intracellular triglyceride content in cells treated with fenofibrate (Figure 2d,e). Overall, the data indicate that treatment of human macrophages with fenofibrate does not reduce the intra-cellular lipid content, but maintains the availability of lipids for the survival benefit of intracellular *Mtb*.

### 4.3. Treatment with Fenofibrate Increased Mtb’s Intracellular Growth

To understand the effect of fenofibrate on the intracellular infection, survival, and growth of *Mtb*, the number of *Mtb* within macrophages were quantified as CFUs. First, the effect of fenofibrate on the phagocytosis of *Mtb* by human macrophages was studied by infecting macrophages with *Mtb* with or without fenofibrate, which was followed by macrophage lysis. The *Mtb-*containing lysate was then cultured for 21 days, and the CFU was calculated. Our data showed that *Mtb* was capable of infecting human macrophages in both groups (Figure 3a), indicating that fenofibrate does not affect bacterial phagocytosis by macrophages. Second, the effect of fenofibrate on the intracellular survival of *Mtb* was studied in human macrophages by co-incubation with *Mtb* for 2 days. Our data showed macrophages treated with fenofibrate harbored a higher bacterial load (Figure 3b). These findings suggest that treatment with fenofibrate does not affect *Mtb* phagocytosis by macrophages but facilitates the growth of *Mtb* in human macrophages.

### 4.4. The Effect of Fenofibrate on the Expression of Dormant, Intracellular Genes of Mtb

The risk of death due to fenofibrate treatment is only evident after nearly a decade; therefore, we hypothesized that the status of *Mtb* latency may contribute to this poor outcome. Hence, we studied the effect of fenofibrate on latent *Mtb* by studying the expression of three dormant genes. Transcriptional analysis was carried out for *icl1* (Rv0467), *tgs1* (Rv3130), and *devR* (Rv3133) by qRT-PCR with 16S rRNA as an internal control. The data showed that the expression of the *icl1*, *tgs1*, and *devR* genes in non-replicating condition was significantly higher in the fenofibrate treatment group than in other three groups (Figure 4).

## 5. Discussion

Our data showed that the risk of death was significantly higher in TB patients treated with fenofibrate. This correlation was even more compelling when subjects were stratified by treatment duration. Because of the rigorous TB reporting system in Taiwan, directed by the Taiwan Centers for Disease Control (CDC), every new active TB case has to be reported within seven days of diagnosis [19]. In addition, it is mandatory for all citizens to be enrolled into the National Health Insurance (NHI) plan, which covers nearly the whole population; therefore, the selection bias due to loss of follow-up was minimized. According to the NHI guidance, prescription of lipid-lowering drugs (e.g., fenofibrate) requires standardized clinical and laboratory assessments [20]. Therefore, the data we used in this study, such as the diagnosis of TB, the prescription of fenofibrate, and the outcomes, are highly reliable. From the database, 2612 patients who were prescribed fenofibrate and 10,448 matched controls in the year 2000 were also studied. However, a similar risk of developing active TB during the 11-year follow-up period was observed (data not shown). Hence, these results suggest that patients treated with fenofibrate did not have an increased risk of reactivation of TB; however, TB patients undergoing prolonged fenofibrate treatment have an increased risk of death. Despite this information, no specific cause of death can be attributed to the use of fenofibrate, making additional analyses with the same dataset difficult. In a separate study, we utilized an in vitro human macrophage system and found that treatment with fenofibrate did not reduce the intracellular lipid content, but tended towards an increase in triglyceride content. Concomitantly, the intracellular growth of *Mtb* was significantly enhanced by treatment with fenofibrate. We assume that even though the intracellular triglyceride content (a major nutrient for the growth of mycobacteria in cells) did not significantly increase with fenofibrate treatment, this small but non-significant increase could be sufficient to support a higher bacterial burden in the macrophages.

The interplay between mycobacterial infection and plasma-lipid content can be complex, partly because of the life cycle of *Mtb,* and the immunologic response to infection [4,21]. Thus, previous data revealed conflicting results in determining whether hyperlipidemia is a foe or a friend of *Mtb* [22,23,24,25]. It has been shown that high serum cholesterol may have a protective effect against respiratory infections, and hypocholesterolemia correlates with susceptibility to TB disease [26,27]. By contrast, a population-based cohort study showed that the use of statins is associated with a lower risk of *Mtb* infection [28]. However, *Mtb* is able to rapidly enter a dormant or non-replicating state, or latent infection, when facing harsh microenvironments. Although host lipids are essential energy sources for mycobacteria during latent state, it is the intracellular lipids accessible to *Mtb* that allow them to persist in the host cells. Therefore, reduction of plasma lipids may not be fully effective at controlling latent TB. In fact, our previous study showed that another lipid-lowering agent, ezetimibe, known to inhibit cholesterol uptake into cells, was associated with a lower risk of latent TB in patients with diabetes [29,30,31]. Our current study shows that fenofibrate not only sustains the availability of intracellular lipids but also enhances dormant genes known to involved in lipid metabolism. Therefore, we hypothesized that prolonged use of fenofibrate may lead to a subclinical *Mtb* activity which eventually results in a higher mortality risk in patients with TB.

The strength of this study is the use of a nationwide database covering >99% of the Taiwanese population, with different risks of death and comparison cohorts. To our knowledge, this is the first study involving in vitro, clinical, and epidemiological data to support the hypothesis that fenofibrate may enhance the intracellular activity and adverse outcome of TB. Despite this strength, there are also several limitations to our study. First, we are aware that the causes of death were similar in each cohort, which does not address how chronic immunologic responses and risks of cardiovascular events each play a role in long-term use of fenofibrate. Second, anti-TB drugs or fenofibrate may cause varying degrees of drug-induced liver injury after a few weeks from start of treatment [32,33]. This study cannot rule out drug–drug interactions and confounding factors affecting early death. Third, intracellular bacterial concentration and survival in vitro does not ideally mimic *Mtb* growth patterns within a host. Fourth, as PPARα is considered the drug target of fenofibrate, it has a wide range of effects on a variety of immune functions [34]. It is indeed possible that mechanisms other than lipid availability may affect the survival of intracellular *Mtb*. Nevertheless, our findings provide important insights into the different aspects of the effects of fenofibrate on *Mtb* infection.

In conclusion, we showed that fenofibrate enhances *Mtb* intracellular growth (Figure 5). Lipid analysis confirmed that fenofibrate does not reduce lipid content in human macrophages as nutrient-rich reservoirs, but upregulates the expression of lipid metabolism-related genes in *Mtb*. According to data from a population database, longer use of fenofibrate in patients with TB is associated with a higher risk of mortality.

## Figures and Tables

**Figure 1 jcm-09-00337-f001:**
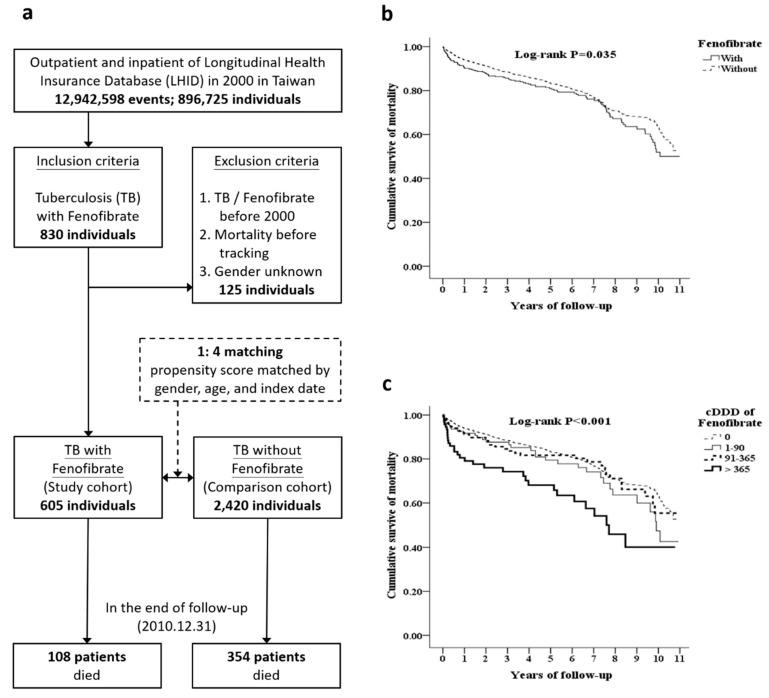
Outcome of tuberculosis (TB) patients on fenofibrate therapy from the National Health Insurance Research Database in Taiwan. (**a**) Flowchart of the study population. (**b**) Kaplan–Meier analysis of the cumulative risk of mortality among the study and comparison cohorts using the log-rank test. (**c**) Kaplan–Meier analysis of the cumulative survival and mortality among TB patients stratified by cumulative defined daily dose (cDDD)of fenofibrate using the log-rank test; 1–90 versus 0 cDDDs; log-rank *p* = 0.245; 91–365 versus 0 cDDDs; log-rank *p* = 0.441; >365 versus 0 cDDDs; log-rank *p* < 0.001.

**Figure 2 jcm-09-00337-f002:**
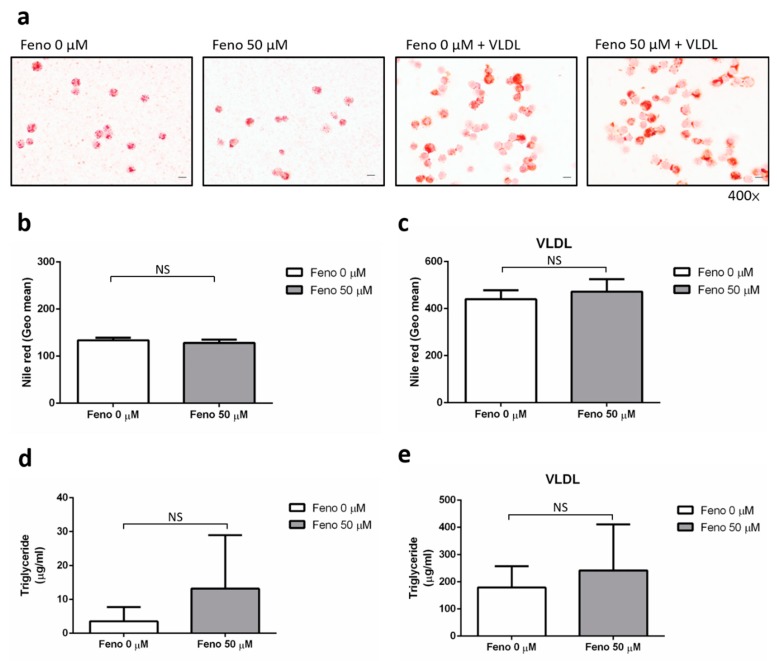
The effect of fenofibrate with/without very low-density lipoprotein (VLDL) on lipid accumulation in human macrophages. (**a**) Human macrophages were stained with Oil Red O and observed by light microscopy on day 2 (400× magnification). (**b**,**c**) For detection of lipid droplets, human macrophages, treated with/without fenofibrate or VLDL, were infected with mycobacteria, stained with Nile Red, and analyzed by flow cytometry (*n* = 15). Lipid accumulation represented as the geometric mean. (**d**,**e**) Quantification of intracellular triglycerides in human macrophages (*n* = 3). Scale bars represent 10 μm. Data are shown as means ± SEMs and were analyzed using an unpaired *t-*test. NS: Not Significant.

**Figure 3 jcm-09-00337-f003:**
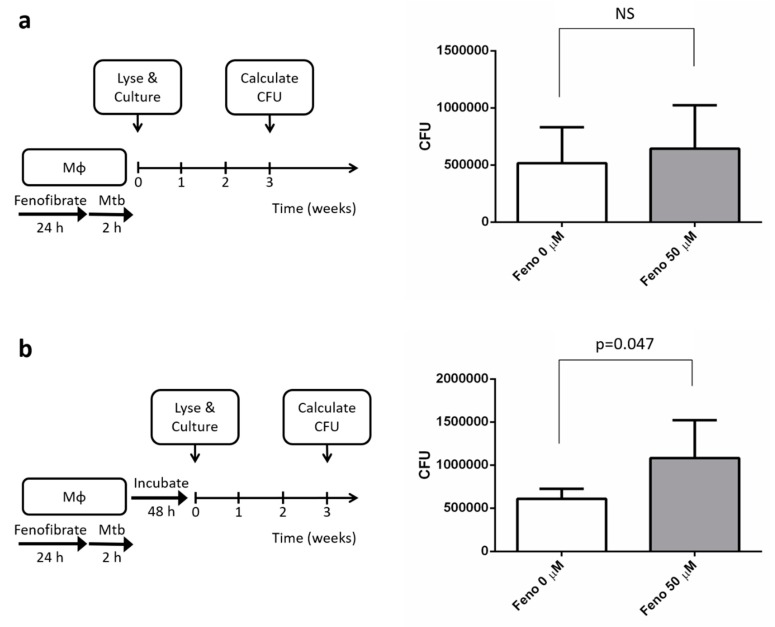
The effect of fenofibrate treatment on the survival and growth of *Mycobacterium tuberculosis* (*Mtb*) in human macrophages (Mφ). (**a**) Human macrophages were treated with fenofibrate (50 μM) for 24 h, infected with *Mtb* (multiplicity of infection, MOI = 5) for 2 h, and then lysed. The lysates were cultured for *Mtb* on 7H11 agar for 3 weeks, after which the numbers of colony-forming units (CFUs) were calculated. (**b**) Human macrophages were treated with fenofibrate (50 μM), infected with *Mtb*, and then washed to remove unbound mycobacteria. After 48 h of incubation, the survival of *Mtb* in human macrophages was determined by *Mtb* culture and CFU determination. Fenofibrate significantly promoted the intracellular survival of *Mtb.* Data are shown as means ± SEMs (*n* = 5 per group), and statistical difference was analyzed with the Mann–Whitney test. NS: Not Significant.

**Figure 4 jcm-09-00337-f004:**
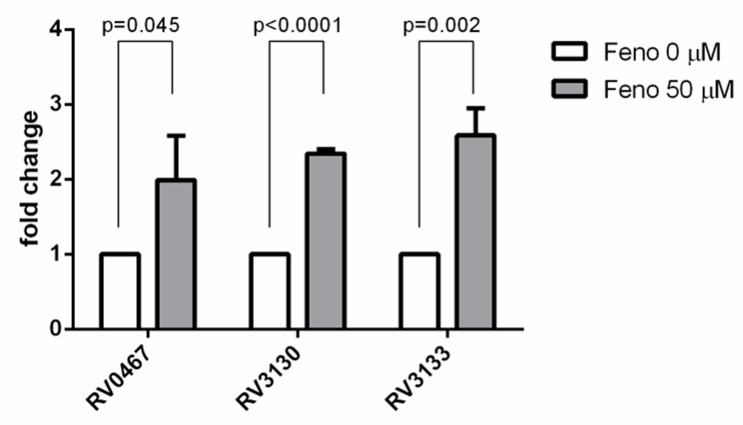
The effect of fenofibrate treatment on the transcriptional response of *Mycobacterium tuberculosis (Mtb)* contained within human macrophages. The mRNA expression levels of Rv0467, Rv3130, and Rv3133 were analyzed on day 2. Data are shown as means ± SEMs, and statistical difference was analyzed using an unpaired *t-*test.

**Figure 5 jcm-09-00337-f005:**
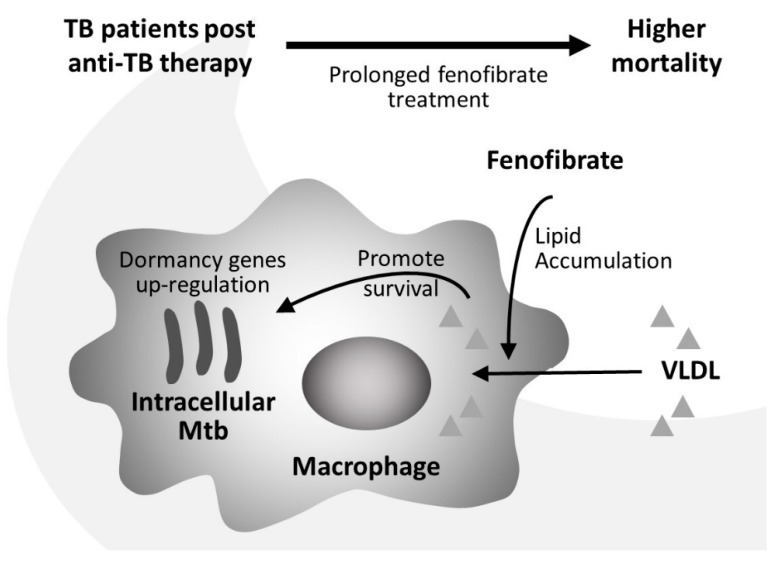
A schematic representation of the proposed mechanism for how fenofibrate, a lipid-lowering drug, can increase intracellular *Mycobacterial tuberculosis* (*Mtb*) viability. Fenofibrate and very low-density lipoprotein (VLDL) induced the formation of foamy macrophages. Next, fenofibrate and VLDL enhanced the survival and growth of *Mtb* in human macrophages. In parallel, fenofibrate and VLDL upregulated the expression of the *icl1*, *tgs1*, and *devR* genes, which could bear relevance to the lipid metabolism and survival of *Mtb* in non-replicating persistent/latent infections. Together, these data indicated that hyperlipidemia with fibrate may increase lipid accumulation in human macrophages as nutrient-rich reservoirs for *Mtb*, which may result in subclinical TB disease activity. Eventually, long-term fenofibrate treatment probably presented higher risk of mortality in patients with post-active TB infection.

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
