# Peer review of "Fenofibrate Facilitates Post-Active Tuberculosis Infection in Macrophages and is Associated with Higher Mortality in Patients under Long-Term Treatment"

_jcm, 2020, doi:10.3390/jcm9020337_

Round 1

Reviewer 1 Report

This manuscript described the association of fenofibrate, a lipid- lowering drug, with mortality in patients after tuberculosis infection. Through a large clinical database analysis with more than 2000 patients, the authors concluded that long-term treatment with fenofibrate in TB patients is associated with a higher mortality.

The authors also used several in vitro methods to further prove that fenofibrate can enhance intracellular growth of mycobacterium tuberculosis (Mtb) as well as upregulate the expression of intracellular Mtb dormant genes. This discovery would be benefit to clinical usage of drugs in TB patients. Some particular comments should be considered.

One major adverse effect of anti-TB drugs including isoniazid and rifampin is hepatotoxicity, which could lead to the discontinuation of drugs for therapy. Is there any association of fenofibrate with hepatotoxicity in TB patients? Is there any drug-drug interaction between fenofibrate and anti-TB drugs? Can such interaction decrease the efficacy or increase the toxicity of anti-TB drugs which further lead to the mortality?

The authors believe that fenofibrate is associated with higher mortality in TB patients. And the cause of death they showed includes tuberculosis and other disease may not be directly related to Mtb infection like cancer. However, the authors provide some mechanism that only related to Mtb. It is hard to correlate with this mechanism to mortality data. One suggestion is to remove the death cause that is not related to TB infection.

The authors checked the VLDL level in macrophage after fenofibrate treatment. It is known that fenofibrate also decrease LDL level. How is the effect of fenofibrate on LDL level? Another question is that this manuscript is already published in European Respiratory Journal. https://erj.ersjournals.com/content/54/suppl_63/PA2944.article-info. Is it only an abstract?

Author Response

We thank the reviewer for the thoughtful recommendations. We answer specific questions and comments below.

# 1

Reviewer: Any association of fenofibrate with hepatotoxicity in TB patients treated with anti-TB drugs? Any drug-drug interaction between fenofibrate and anti-TB drugs? Can such interaction further lead to the mortality?

Author: Indeed, the occurrence of hepatitis during the treatment of TB is an important issue. Anti-TB drugs and fenofibrate may cause abnormal liver function and drug-induced liver injury in a small number of patients. It usually occurs several weeks after using these drugs and even causes acute liver failure. This study cannot rule out drug-drug interactions and confounding factors affecting early death. We added some explanations “Second, anti-TB drugs or fenofibrate may cause varying degrees of drug-induced liver injury…confounding factors affecting early death.” at the 'limitation' paragraph in Page 16, Line 310-312. And Ref 32, Ref 33.

# 2

Reviewer: For the correlation with cause of higher mortality and mechanism related to Mtb, suggest to remove the death cause that is not related to TB infection.

Author: We agree. The purpose of this study was to investigate the effect of fenofibrate on the intracellular survival of Mtb, which is one of many reasons that may increase the mortality after TB infection. Describing too many causes unrelated TB infections will blur the focus. Therefore, we revised 'results' paragraph in Page 9, Line 178-179.

# 3
Reviewer: The authors checked the VLDL level in macrophage, but not LDL? How is the effect of fenofibrate on LDL level?

Author: VLDL is a good source of triglyceride. LDL has a lot of cholesterols, which will make experimental variables difficult to control. We used added VLDL to make a simulation of human macrophages in a hyperlipidemia environment. But we didn’t use LDL or check its intracellular level.

# 4

Reviewer: Is this manuscript is already published in ERJ?

Author: Some of the epidemiology results have been previously published in abstract form in the ERS International Congress 2019, Madrid, Spain. We revised the ‘abstract’ and delete “(log-rank test, p = 0.035)” and “(p < 0.001)” in Page 3, Line 41 and 42.

Reviewer 2 Report

Liu et al in their paper investigate long-term effects of fenofibrate treatment in TB patient,They also acess the intracellular viability of Mtb in human macrophages in culture conditions. In general the authors can and should do more to make the study more relevant and interesting to the general audience.

I also have a few specific comments:

Line 58 has typographical error - ntroduction

Line 129 space between 37 and °C

Figure 4 legend: consider changing "contained with human macrophages" to "contained within human macrophages"

Line 336: statistically should be statistical

In general the paper can benefit from a discussion on the mechanism of action of fenofibrate. fenofibric acid activates peroxisome proliferator activated receptor α (PPARα) thereby increasing lipolysis and elimination of triglyceride-rich particles from plasma. The authors should discuss the drug's target and the effects it has on the intracellular levels of various lipids. The authors have simply shown the affects of fenofibrate on cells infected with M.tb. Are the observed effects due to off target effects of the drug or because of target engagement? A simple test could be to include a control with external supplementation of triglycerides/LDLs

The paper also does a poor job of describing the role of redox related proteins in Mtb during dormancy and reactivation. In particular I suggest the authors PMID: 23823726, the paper can benefit from a further study of transcription factors like Rv0081. The authors also do not cite any references for the three genes that they selected for their RT PCR assays. Furthermore, I suggest that the authors do a better job in trying to explain their results using a schematic representation of the mechanism

Author Response

We thank the reviewer for the thoughtful recommendations. We answer specific questions and comments below.

# 1

Reviewer: Typographical error in Line 58, Line 129, Figure 4 legend, and Line 336.

Author: We have corrected all of them.

# 2

Reviewer: The authors should describe the fenofibrate's target and the effects it has on the intracellular levels of lipids.

Author: We revised and added a description of the fenofibrate mechanism “By the activation of peroxisome proliferator activated receptor α (PPARα)…hyperlipidemia [13-15].”in the 'introduction' paragraph in Page 4, Line 75-79. And Ref 11, Ref 12, Ref 15.

# 3
Reviewer: The authors have simply shown the affects of fenofibrate on cells infected with Mtb. Are the observed effects due to off target effects of the drug or because of target engagement?

Author: Thank you for your question. In this manuscript, we do not aim to study the drug target of fenofibrate. However, in our earlier studies fenofibrate had direct growth inhibition effects on cultured Mtb, in contrast to its role in macrophage (see figure below), and these effects could also partly be reversed by the addition of triglycerides (VLDL). Besides, as PPAR-α is considered to be the drug target of fenofibrate, it has a wide range of effects on a variety of immune functions. It is indeed possible that mechanisms other than triglyceride may affect the survival of intracellular Mtb. We added explanation “Fourth, as PPARα is considered the drug target of fenofibrate...affect the survival of intracellular Mtb.” in the ‘limitation’ section in Page 16, Line 314-316. And Ref 34.

# 4

Reviewer: The authors also do not cite any references for the three genes that they selected for their RT PCR assays.

Author: We have cited 3 reference papers in the 'Methods' paragraph, Page 8, Line 157, Ref 16, Ref 17, and Ref 18.

# 5

Reviewer: Trying to explain the results using a schematic representation of the mechanism.

Author: We have added a new Figure 5 to create a schematic diagram to explain the possible mechanisms found in this study, in the ‘conclusion’ section in Page 16, Line 320. The new Figure is in Page 17, Line 325. The figure legend is in Page 17, Line 326-336.

Round 2

Reviewer 1 Report

NA